# Sand Play for 0–8-Year-Old Children’s Health and Development: A Systematic Review Protocol

**DOI:** 10.3390/ijerph181910112

**Published:** 2021-09-26

**Authors:** Susanna Iivonen, Titta Kettukangas, Anne Soini, Helena Viholainen

**Affiliations:** 1School of Applied Educational Science and Teacher Education, University of Eastern Finland, 80101 Joensuu, Finland; titta.kettukangas@uef.fi; 2Department of Education, University of Jyväskylä, 40014 Jyväskylä, Finland; anne.j.soini@jyu.fi (A.S.); helena.j.k.viholainen@jyu.fi (H.V.)

**Keywords:** sand play, early childhood, health, development, protocol, systematic review

## Abstract

Sand play may be a significant determinant of health and development in early childhood, but systematically synthesised evidence is absent in the literature. The main objective of this study was to present a planned methodology to systematically review, and synthesise, the evidence regarding sand play and its associations with 0–8-year-old children’s health and development. The systematic review following the Preferred Reporting Items for Systematic Reviews and Meta-Analyses Protocols statement was registered to PROSPERO (CRD42021253852). Literature searches will be conducted using information from eight electronic databases. Studies will be included when participating children were aged 0–8 years, settings provided children with exposure to sand environments and/or materials, and child-level outcomes related to physical, cognitive, and/or social–emotional health and development. The search results will be imported to software; duplicates will be removed; and independent double screening, and study quality assessments using appropriate tools, will be conducted. Synthesis without meta-analysis will be conducted for quantitative studies similar in exposure, outcome, and content analysis to qualitative studies. Our overall confidence in each review finding will be assessed. The findings of this systematic review can inform policy makers and early childhood education teachers about the associations between sand play and children’s health and development, and its impact in practice.

## 1. Introduction

The World Health Organization’s (WHO) guidelines for young children [1] state that children should have opportunities to participate in a range of developmentally appropriate, safe, and enjoyable play-based physical activities. The guidelines also suggest that children should be physically active for at least 180 min at any intensity throughout the day. Sand play is a potential avenue by which to promote these goals. Its global prevalence is high, and the sandpit is the most common fixed environment for early childhood education ECE [2]. Regardless of geographical, linguistic, or cultural context, one finds sandpits and sandy environments in populated areas wherever one travels, including in home yards, playgrounds, and other leisure spaces for children, families, and groups. In addition, children themselves consider sand play a pleasant activity as it involves manipulation, exploration, and construction with materials to create imaginary worlds [3,4,5].

Sand play is an integral part of physical education (PE) during the early years (0–8 years). Developmental PE involves the study and application of the ways in which children acquire skills in the motor, fitness, cognitive, and affective domains of learning in order to become physically active over their lifetime [6] (p. 8). PE is developmental when the ways in which an individual acquires skills settle in the zone of proximal development [7]. The zone of proximal development describes the skill level where the child can manage with help, but the task is too difficult to manage alone. Playing in the zone of proximal development is sufficiently challenging and pleasurable for a child [7]. From this point of view, challenging sand play can act as a zone of proximal development for motor learning. Further, in the ecological approach to development, the interplay between the actor and the environment is crucial [8]. Thus, individuals perceptions of the physical opportunities afford activity that is in the range of individual capabilities. Therefore, as previous studies have shown, natural materials such as sand may stimulate children engaging in activities that develop movement [9] as well as constructive, and functional, play [10,11].

The concept of sand play can be viewed broadly, as encompassing all different kinds of sand environments and materials including traditional sandboxes, delimited sandy areas of different sizes, and sand materials, both outdoors and indoors, in any geographical or cultural area. According to Pellegrini, Dupuis and Smith [12], play is a central component for a young child, which is why the concept of play can be used to refer to any activity undertaken by a child. The ‘sandbox’ (US and Canada) or ‘sandpit’ (most Commonwealth countries) was previously called a ‘sand table’ or ‘sand garden’, and it originates from a suggestion by Friedrich Froebel in his 1850s work on kindergartens [13]. According to Hart [14], sand play is a natural evolution from the informal play with earth that children all over the world engage in, during early childhood, given that sand is a cheap and hygienic alternative to free play with other materials [15]. In his plan of a model kindergarten, Froebel encouraged the design of a garden that offered children opportunities to come intocontact with natural materials, such as sand [14,15]. The first known use of sand for play was through the so-called ‘sand bergs’ piled in the public parks of Berlin in 1850. At that time, the kindergarten movement in Germany began to include sandpits in their settings [14]. The idea caught on quickly in the United States, where sandpits were mostly mostly in poor neighbourhoods, often alongside women-managed settlement houses for servicing the children of immigrants [14]. Public sand play places spread rapidly in industrialised countries in the early part of the twentieth century, but in the latter part of the century their number declined due to growing concerns over their supposed toxicity due to animal faeces and municipal agencies’ decreased willingness to carry out the, necessary, occasional maintenance [14]. The earliest known sandpits included wheelbarrows, rakes, buckets and spades, and moulds of animals and other shapes [16]. Today, implements are usually limited to smaller tools, and these are made of plastic rather than metal [14].

Why are sandpits, and children’s play in them, popular around the world? Sand is a material with a particle size of between 2 millimetres (1/12 inches) and 0.06 millimetres (1/400 inches) [17]. As a granular material, it can be mounded, poured, and measured when dry. When it is wet, the surface tension of the water causes the grains to stick together, allowing the sand to be moulded, shaped, and carved. Sand is accessible all over the world and, in contrast to other granular materials, it does not readily decompose [17]. The tactile qualities of sand fit well with many goals of ECE. Sand play has been viewed as valuable for its cognitive and social benefits [18]. Piaget’s play levels—functional (e.g., filling containers), constructive (e.g., building sand houses), and dramatic (e.g., making and tasting birthday cakes)—serve as a reference frame for what he called ‘mental complexity’ [19] (p. 108), and are possible for children to achieve with sand play. In addition, while engaged in sand play, children may learn important concepts such as scientific and mathematical principles; for example, those relating to mass and capacity [20]. Jarret, French-Lee, Bulunuz and Bulunuz [21] have suggested that many important physical, cognitive, and social–emotional skills can be learned with sand, including fine motor and gross motor skills, measurement, cooperative building, sharing, and pretending. In addition, sand play is utilised worldwide as a psychotherapy method [22]. In sandplay therapy, a child is provided with a sandbox and helped to make, modify, talk about, and change their thoughts and emotions by using sand, water, miniatures, and their hands [22]. As a nonverbal approach, such therapy is especially effective when working with children experiencing trauma, distress, disabilities, and migration issues. Paediatrician Margaret Lowenfeld is considered the first person to have described sand play as a therapeutic technique; a concept that she, in turn, based on Jungian psychoanalysis [22].

Based on the information presented above, sand play seems to be associated with children’s health and overall development. However, fragmented evidence suggests that such benefits are not fully understood and, therefore, cannot be considered in different contexts. For instance, one study suggested that sand promoted about one fifth of the activity on a playground with a large sand area but that very little of it could be considered moderate-to-vigorous physical activity [23]. The authors of the study, furthermore, noted that ‘by nature, sand was sedentary’ [23] (p. 517) and that, as a surfacing material, sand inhibited physical activity ‘likely because it makes running difficult’ [23] (p. 518). Another study showed that sandpits may have limited children’s physical activity in the ECE outdoor setting [24]. However, the association remained unclear because, when other factors such as the equipment provided to children were taken into account in the analysis, the negative association disappeared [24]. The authors concluded that the benefits of sandpits, and their role in helping children to engage in physical activity, remain unknown and require further study [24]. Parten [25] found, in his extensive observational study, that children played in sandpits more frequently than engaged in any other kind of play activity, but that sand did not promote much social play. Sand play was predominantly a parallel play activity, and cooperative work was rare [25].

In line with the WHO’s [1] guidelines,—and considering the assumed high global prevalence of sand play,—children’s sand play must be explored to ensure that developmentally appropriate, play-based, physical activities for young children exist. Global evidence on children’s sand play, as well as its effects and associations with children’s physical, cognitive, and socio-emotional health and development, should be systematically reviewed. Analysing and synthesizing such data can improve our understanding of what kind of sand play environments and situations promote, or reduce, children’s health and development. The results can then be considered and applied to children in different environments and contexts as well as in future studies. Therefore, the aim of this research is to systematically review the published evidence to:Aggregate studies examining 0–8-year-old children’s sand play;Explore what determinants of child health and development are affected by, or associated with, sand play.

## 2. Material and Methods

This systematic review protocol was registered to the International Prospective Register of Systematic Reviews (CRD42021253852). The protocol is reported following the Preferred Reporting Items for Systematic Reviews and Meta-Analyses Protocols (PRISMA-P) statement [26,27] (see checklist in Appendix A). The forthcoming systematic review will be reported in line with the PRISMA 2020 statement, an updated guideline for reporting systematic reviews [28]. The protocol is described in the following sections.

### 2.1. Eligibility Criteria

Studies will be selected following the (S)PI(E)COS (Study designs, Population, Intervention or Exposure, Comparison, Outcomes, and Setting) framework outlined below (see inclusion and exclusion criteria in Appendix A).

#### 2.1.1. Study Designs

All primary, quantitative and qualitative, research designs will be considered in studies where outcomes were child-level and assessed in a sand play environment. Studies that examine perceptions from parents, educators, practitioners, or children on child-level outcomes using children in a sand play environment will be included.

#### 2.1.2. Participants

Based on the World Health Organization’s [29,30] definition of the age at which early childhood development occurs, children aged 0 to 8 years of age will be included. Typically and atypically developing children with diagnosed chronic disease or disability (for example, asthma, physical disability, or attention deficit hyperactivity disorder) will be included. ECE in many countries coincides with these ages, although this varies by country (e.g., China and Finland 0–6 years, Kenya 3–7 years, Sweden 1–5 years; United States children from birth through to compulsory, state-designated, school age; 5–8 years). Children’s reported mean age, range, or median will be used to decide whether the study is eligible. If a study is conducted in an ECE setting, but no age is reported, it will be considered. Retrospective designs, where outcomes were assessed at the time the children were ≤8, will be included. Regarding longitudinally designed studies in which children were followed at different ages, we included the part of the study in which the children were aged 0–8 years. Likewise, regarding research designs including groups of children of different ages, we included the part of the study in which the children were aged 0–8 years.

#### 2.1.3. Interventions/Exposures and Comparators

In this study, sand play works as a top concept that covers different kinds of sand environments and materials outdoors and indoors. These could include traditional, low-edged sandboxes of different sizes; delimited sandy areas of different sizes; or sand environments and materials built indoors. All geographical and cultural demographics will be accepted. The concept of play is used because it is a central component of life for a young child [12]. Play means all kinds of being, doing, acting, or behaving, etc. by a child occurring in a sand environment or with sand materials. Sand play could be embedded in an intervention program or occurring in ECE, at home, in a neighbourhood, in a park, in an an organisational activity, or as part of a rehabilitation program. Any comparator study was included, as were study designs that did not have a comparator.

#### 2.1.4. Outcomes

All child-level outcomes are of interest to this study. Different aspects of health and development [29,30] will be accepted. These could include physical factors (e.g., physical activity, fundamental motor skills, perceptual-motor skills, fine motor skills, weight status, fitness, infectious conditions, and injuries), cognitive factors (e.g., attention, memory, pattern recognition, executive functions, mathematical skills), or social–emotional factors (e.g., interactions, pro-social behaviour, resilience, feelings of stress, trauma) with respect to children’s health and development.

#### 2.1.5. Settings

There will be no restrictions to the type of setting provided that the above criteria are met.

### 2.2. Information Retrieval

#### 2.2.1. Information Sources

Electronic databases of articles relating to (1) thebehavioural sciences, (2) forestry and environmental science, (3) health and medicine, (4) the humanities and arts, (5) multidisciplinary studies, (6) mathematical sciences, and (7) social sciences were screened for possible information sources. Experimental searches were performed in 14 databases to examine their relevancy, and geographical and cultural representation for the systematic review. Of these, the following seven were selected:CINAHL (EBSCO)EBSCOhost Academic Search PremierERIC (ProQuest)Medline (ProQuest)Scopus (Elsevier)SPORTDiscus with Full Text (EBSCO)PsycINFO (EBSCO)

Open Grey (http://www.opengrey.eu, accessed on 27 November 2020), DART-Europe E-theses Portal (http://www.dart-europe.eu/About/info.php, accessed on 27 November 2020), Directory of Open Access Journals (https://doaj.org, accessed on 27 November 2020), and Google Scholar (https://scholar.google.com, accessed on 27 November 2020) have been checked for possible sources of grey literature, such as dissertations and research reports. Google Scholar will be searched, and the first 10 pages will be screened in the systematic review. To ensure literature saturation, the reference lists of included studies and relevant reviews identified through the search, will be scanned. Websites of relevant organisations and other groups involved in children’s sand play-related environments will also be searched. In addition, relevant organisations, practitioners, and researchers in the field will be contacted to obtain information.

#### 2.2.2. Search Strategy and Study Records

Various database indexing terms relevant to the concepts of ‘children’ and ‘sand play’ will be examined. The specific search strategies for each database will be created by an information specialist librarian. The search will be limited to humans, 0–8 years old, and peer-reviewed English publications, where possible. In addition, the search will be limited to articles, conference proceedings, research reports, and dissertations. No date limits will be set. A draft ERIC search strategy is included in Appendix A.

Literature search results will be imported to Refworks 3.0 (Proquest) software, which will be used to de-duplicate references, upload abstracts and full texts, as well as facilitate collaboration among reviewers during the study selection process. Step one (title and abstract screening) and step two assessments (full text screening), against inclusion and exclusion criteria in the Refworks software, will be developed before the formal screening process.

#### 2.2.3. Selection Process and Data Extraction

One reviewer will remove duplicates. Titles and abstracts will be screened by one reviewer (S.I., T.K.) and 15% of the titles and abstracts will be screened independently in duplicate. Full text documents will be screened independently (S.I., T.K.) in duplicate. Additional information will be sought from the study’s authors, where necessary, to resolve questions about eligibility. In cases where reviewers disagree during any step of the screening process, discussion with a third reviewer (A.S., H.V.) will be used to resolve disagreement. Where there are multiple publications for the same study, all publications will be combined and reported as a single study.

#### 2.2.4. Data Collection Process

Data will be extracted from included studies using a data extraction template, which will be piloted before the formal extraction process (see Appendix A). In piloting the template, two reviewers (S.I., A.S.) will independently extract data from two quantitative studies, and two reviewers (T.K., H.V.) will independantly extract data from two qualitative studies. Reviewers will compare each others’ entries, and make necessary adjustments to the template to achieve sufficient consistency. In the formal extraction process, data from each study will be extracted by one reviewer and cross-checked by another reviewer. In cases where data might be missing, or additional information needed for the eligible studies, the study’s authors will be contacted to provide the information. An email will be send to the corresponding author asking for the required information. A second email attempt to obtain missing information will be sent three weeks after the initial email and a third email attempt one week after the second attempt. If no response is received within six weeks after the initial contact, the study may be excluded from the systematic review. The following data will be extracted:Study details.
○Quantitative: author(s), year, geographical location (i.e., country), study design (e.g., longitudinal, cross sectional).○Qualitative: author(s), year, geographical location (i.e., country), study design (e.g., ethnographic, narrative research, historical, case study, phenomenology), and study aims.Participants: age, sample size, gender, and background information.Intervention/exposure: description, duration, and follow-up.Data collection methods.
○Quantitative: assessment and analysis methods (e.g., accelerometers, direct observation), and time point of assessment.○Qualitative: methodology (e.g., interviews, focus groups), and method of analysis.Findings.
○Quantitative: outcome findings (outcome unit, effect estimates, standard deviations, confidence intervals, direction of effect, statistical significance, etc.) and conclusions drawn from the findings by the authors of the study.○Qualitative: summary of findings and conclusions drawn from the findings by the authors of the study.

### 2.3. Quality Assessment of Included Studies

Several methodological tools have been developed for assessing the quality (or risk of bias) of studies included in systematic reviews for medical research, making them not directly applicable to the evaluation of behavioural sciences [31]. Farrah, Young, Tunis, and Zhao [32] have argued that there is a lack of knowledge on which tools are the most rigorous and practical for different purposes and study designs, including quantitative experimental (e.g., randomised controlled trials and non-randomised interventional studies), observational (e.g., cohort, case-control, cross-sectional studies) and qualitative studies. Further, the single assessment tool may not adequately assess the quality in a way particular to the different study designs included in a systematic review. However, use of multiple tools in a systematic review may be problematic due to different tools having different rating systems, making the comparability between qualities difficult [32]. At least three domains should be covered: appropriate (1) selection of participants (2) measurement of variables, and (3) control of confounding variables, and consideration of design-specific biases [33].

The use of three different quality assessment tools will be prepared. The quality of quantitative interventional studies will be assessed using the Effective Public Health Practice Project (EPHPP) Quality Assessment Tool [34]. Descriptive cross-sectional studies will be assessed using the National Heart, Lung and Blood Institute’s (NHLBI) Quality Assessment Tool for Observational Cohort and Cross-Sectional Studies [35]; qualitative studies will be assessed using the Standards for Reporting Qualitative Research (SRQR) [36].

Some modifications will be made to the tools to make them suitable for this systematic review. For instance, if there is insufficient detail in a descriptive cross-sectional study paper such that the reviewers are unable to determine if the answer is ‘yes’ or ‘no’ to an NHLBI criterion, the original study’s authors will be contacted via email. If the necessary information is not received within a month of contacting the author of the paper, the answer ‘no’ will be assumed for that criterion. In assessing the quality of qualitative studies, a study will be excluded if the paper does not provide answers to the SRQR items from 1 to 19. The justification for the exclusion of a study, based on these limitations, is that the results would not be reliable if, for example, the study was not described with sufficient accuracy to ensure the reliability of the methods used. To facilitate comparisons of the quality of the included quantitative studies, evaluated with different assessment tools, an extra total scoring system that converts the total quality score into a percentage will be added to each tool (EPHPP, NHLBI).

The quality of all included studies (S.I., T.K., A.S., H.V.) will be assessed independently in duplicate. In instances of disagreement, a third reviewer will be brought in to resolve differing views. A draft of the quality assessment tools, with modifications highlighted, can be found in Appendix A.

### 2.4. Data Synthesis

A systematic narrative synthesis according to the guidance [37], and synthesis approaches that allow heterogeneity of the included studies in terms of designs, participant groups, exposure types, assesment and analysis methods, as well as reported findings (e.g., effect estimates) will be implemented. Tabulation will be used to describe the characteristics (design, geographical location, participants, intervention/exposure, assessment and analysis methods, investigated outcomes) of all included studies. Separate tables will be constructed for quantitative and qualitative studies. If a quantitative part of a mixed-method study is eligible for the systematic review, that part will be tabulated with quantitative studies, and if the eligible part is qualitative, it will be tabulated with qualitative studies. If both quantitative and qualitative parts of a mixed-methods study are eligible, each will be tabulated accordingly in both tables and a note that it is a single study will be made. A systematic narrative synthesis will also be provided, highlighting the different study designs, geographical locations, participant characteristics, sand play exposure types, and investigated child-level outcomes of the eligible studies. Synthesis of quantitative studies will be conducted by two reviewers (S.I., A.S.). Synthesis of qualitative studies will also be conducted by two reviewers (T.K., H.V.) independently, in duplicate, and cross-checked. Disagreement will be resolved through discussion with a third reviewer. In instances where data may be missing or additional information required for the eligible studies, the study authors will be contacted to provide the relevant information. Conclusions drawn will be based on better-quality evidence. The findings of the forthcoming systematic review will be based on the planned syntheses of eligible quantitative, qualitative, and mixed-method studies described in the following subsections.

#### 2.4.1. Synthesis of Quantitative Studies

First, quantitative studies will be analyzed to distinguish the different sand play intervention/exposure types (e.g., sand play in ECE) of the eligible studies. Second, within each exposure type, the outcomes assessed in the studies will be examined. If more than one quantitative study reports data on the same outcome (e.g., physical activity), we will consider conducting a Synthesis Without meta-analysis (SWiM), based on vote counting of the direction of effect [38]. The reported direction of effect, or association (positive, negative or no effect/association), for each of the eligible studies within the same exposure type (e.g., sand play in ECE) on a particular outcome (e.g., physical activity) will be coded as follows: ↑ = positive effect/association; ↓ = negative effect/association; → = no effect/association. Findings of each outcome (e.g., physical activity) per eligible study within the same exposure type (e.g., sand play in ECE) will be then tabulated alongside key study characteristics such as study design; sample size; effect estimates, if available; effect direction (↑/↓/→), and quality rating. Using these tables, vote counting will be conducted to compare the number of studies showing positive effects/associations with the number of studies showing negative effects/associations for that particular outcome. This will address the question as to whether there is any evidence of an effect or association of the intervention/exposure type on that particular outcome. To illustrate the vote counting results on the particular outcome, harvest plots will be provided. In the harvest plot, a subset of studies with similar intervention/exposure type will be categorised based on their effect directions (studies with ↑; studies with ↓; studies with →) and grouped together. In the harvest plot, each study will be represented by a bar positioned according to its categorisation. The height of a bar will depict the quality judgement of a study (e.g., tall = high quality; medium = limitations; short = low quality) and the alphabet beneath the bar will represent the studies. In addition to tables and harvest plots, as described above, a narrative synthesis will be provided to report on findings grouped by outcome and intervention/exposure (e.g., sand play in ECE on physical activity), as described above.

#### 2.4.2. Synthesis of Qualitative Studies

With qualitative studies, a translation of data using thematic and content analysis [37] will be conducted. First, primary themes or concepts reported in the studies will be examined to explore similarities and/or differences between different studies to group them into higher-order themes (e.g., sand play therapy; sand play in a specific cultural context). Second, within the higher-order themes, a content analysis will be considered to compress many words of text reported in the included studies into fewer content categories based on explicit rules. If the type and number of data allow coding rules that allow the findings reported in the eligible studies to be categorised according to the child’s development domains (physical, cognitive and social–emotional) and/or particular aspect of a development domain (e.g., physical domain—fine motor skills; social–emotional domain—social interaction skills), further categories will be developed. Higher-order themes and content categories linked to each will be tabulated alongside key study characteristics and findings. If a single study reports finding of multiple themes, it will be treated accordingly, and a note will be added to the table. In addition, a narrative synthesis will be provided to report findings grouped by higher-order themes and content categories, as described above.

#### 2.4.3. Synthesis of Mixed-Methods Studies

The eligible quantitative data of mixed-method studies will be included in the synthesis of quantitative studies; eligible qualitative data will be included in the synthesis of qualitative studies, as described in the previous subsections. In cases where the data of a single mixed-method study are eligible for both syntheses (e.g., quantitative part: association of sand play in a specific cultural context on children’s physical activity; qualitative part: perceptions of parents of children’s sand play activity in a specific cultural context), data will be used in both syntheses and a notification that it is a single study will be made.

#### 2.4.4. Summary Syntheses

If possible, the findings of quantitative and qualitative studies will be summarised by combining them into a single logic model, which will be developed by the reviewers (S.I., T.K., A.S., H.V.). The aim of the model will be to show different exposure types (e.g., sand play in ECE; sand play in public areas; sand play in a specific cultural context; sand play for therapeutic purposes) and data types (quantitative and qualitative) and their associations with child-level outcomes of health and development (e.g., potential pathway—how sand play in ECE can benefit children’s physical activity; potential pathway—how sand play in a specific geographical and cultural context might be associated with children’s social interaction).

### 2.5. Strength of the Body of Evidence

The Confidence in Evidence from Reviews of Qualitative research (GRADE-CERQual) approach will be utilised to assess the overall confidence in each review finding based on consideration of the four components [39]: (a) methodological limitations of the individual studies contributing to each review finding will be assessed utilising the average of study quality assessment percentage scores, as described above [40]; (b) the coherence of each review finding will be assessed by exploring how clearly and cogently the data fit with the review finding (i.e., the extent to which the data support the phenomenon of interest, the clarity of the data) [41]; (c) the adequacy of the data that supported a review finding will be assessed in terms of the thickness of data (the number of studies, the number of participants, cultural and geographical coverage) [42]; and (d) the relevance will be assessed based on the extent to which the review finding is applicable to the context (setting, population) specified in the second review question [43]. Each of these four components will be assessed as having minor, moderate, or substantial concerns regarding the specific component. Based on an overall assessment of the four components for each review finding, they will be assessed as high, moderate, or low [44].

## 3. Discussion

This study protocol presented a planned methodology for a systematic review of studies examining sand play in 0–8-year-old children and child-level health and development determinants that were found in them. In the systematic review, the narrative synthesis will be conducted to report findings and, where possible, syntheses without meta-analyses will be performed to provide a more rigorous evidence base for sand play. Limitations of the studies that will be included in the systematic review will be reported and taken into account during synthesis and the interpretation of the findings. The systematic review will have a number of limitations, such as heterogeneity of the eligible studies, that will be reflected in the synthesis approaches, the quality assessment tools and assessment of the strength of the body of evidence, which may be limited. In addition, another limitation will be that data search will be limited to English-language publications, which may exclude important research information reported in non-English languages. However, to reduce this limitation, preliminary searches have been conducted in 14 different electronic databases and three different databases of gray literature. Based on this, data sources will be selected in such a way that research from different cultures and linguistic areas will be as well represented as possible in the forthcoming systematic review. A strength will be that new and rigorously developed procedures will be applied for a systematic description, analysis and synthesis of behavioural science research data, which is still performed relatively rarely [31]. If any changes are made to this protocol, these will be described in the published systematic review.

Correlates, determinants and outcomes of sand environments and materials, without delimiting any geographical or cultural area on children’s health and development, will be identified, analysed and synthesised in this systematic review. Most optimally, it will be possible to demonstrate the beneficial and detrimental effects and associations that sand play in different exposure environments has been found to have on children. Those effects and associations can be then considered, utilised or avoided for a variety of uses and contexts with children. For example, beneficial practices identified in sand play therapy can potentially be introduced and applied in the early years of PE to provide play environments that support the favourable development and health of children with special needs: or, if the sand play toys and equipment used in a particular culture have been found to limit the diversity of children’s motor and social skills, the toys and equipment may be altered. Similarly, if the sand play objects used by children in other circumstances have been shown to be associated with a greater number of motor skills and better social skills, such objects may be provided for use by children in other contexts as well.

## 4. Conclusions

The systematic review based on this study protocol will be valuable because of the way it will show quantitative and qualitative evidence on sand play and its effects and associations with children’s physical, cognitive and social–emotional health and development outcomes. Furthermore, this systematic review will show that themes that have rarely been studied in the context of sand play might nevertheless have importance to children’s development based on earlier research literature. These findings will have implications for policymakers, ECE managers, ECE and PE teachers, architects, landscape architects and other stakeholders in the field. Sand play has a long history and fascinates children around the world. It is important to find out its effects on children’s health and development so that children will continue to enjoy it, experience inclusion and equality, and make progress in their overall development.

## Data Availability

The data included in this review protocol are available in the supplementary material and recorded and maintained as a permanent record on an open access electronic database appended with the links to subsequent systematic review publication in the International Prospective Register of Systematic Reviews (CRD42021253852). Any data that will not be available in the PROSPERO record or in the forthcoming published systematic review, such as reviewed studies that were not publicly available, can be obtained from the corresponding author.

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
