# Peer review of "Sand Play for 0–8-Year-Old Children’s Health and Development: A Systematic Review Protocol"

_ijerph, 2021, doi:10.3390/ijerph181910112_

Round 1
Reviewer 1 Report
The study is relevant, original, but requires some improvements, as it reports in the method and conclusion to quantitative, qualitative and mixed method studies, but only the qualitative analysis is well designed, with no details on quantitative studies or on those of mixed method, which need to be improved substantially in the text.
Reviewer 2 Report
I would consider it appropriate to explain why non-English publications will be excluded. It is possible that this exclusion leads to the selection of only some geographical areas, which in itself would not be negative: the introduction refers to European and American authors and educational situations, if this choice is to be maintained it must be clearly stated. Otherwise I would consider it useful to specify if and how the variables linked to different cultural contexts will be included in the analysis. If a preliminary search has been made highlighting the scarcity / absence of literature in other contexts besides the Anglophone it must be declared.
Round 2
Reviewer 1 Report
The authors made the requested corrections and I recommend ACCEPT the corrected version of the article for publication.